# Bionic Synthesis of Mussel-like Adhesive L-DMA and Its Effects on Asphalt Properties

**DOI:** 10.3390/ma15155351

**Published:** 2022-08-03

**Authors:** Jinyi Wu, Quantao Liu, Shaopeng Wu

**Affiliations:** State Key Laboratory of Silicate Materials for Architectures, Wuhan University of Technology, Wuhan 430070, China; jinyi_wu@whut.edu.cn (J.W.); wusp@whut.edu.cn (S.W.)

**Keywords:** L-DMA, mussel bionic materials, modified asphalt, modification mechanism, rheological properties

## Abstract

Cracks are inevitable during the service life of asphalt pavement and the water at the fracture surfaces tends to cause the grouting materials to fail. Studies have shown that the catechol groups in adhesion proteins secreted by mussels can produce strong adhesion performance in the water. In this paper, the mussel-like adhesive L-Dopa Methacrylic anhydride (L-DMA) was prepared based on the concept of bionic design and used to improve the properties of asphalt. By using Fourier-transform infrared spectroscopy (FTIR) and Thermogravimetric analysis (TGA), the thermal stability and structural composition of L-DMA were investigated. Then, the rheological and low-temperature properties of L-DMA-modified asphalt were investigated using the dynamic shear rheological (DSR) test and bending beam rheological (BBR) test. Moreover, the modification mechanism was explored by FTIR. It was found that L-DMA can be effectively synthesized and has good thermal stability. The incorporation of L-DMA increases the composite modulus, viscosity, creep recovery rate and rutting factor of asphalt binder, resulting in an enhancement of its high-temperature performance. At a high L-DMA content of 10%, the low-temperature performance of the modified asphalt was enhanced. The modification of L-DMA to asphalt is mainly a physical process. Hydrogen bonds and conjugated systems generated by the introduction of catechol groups enhance the adhesion properties of asphalt. In general, L-DMA improves the properties of asphalt and theoretically can improve the water resistance of asphalt, which will be explored in future research.

## 1. Introduction

The road network in China has significantly improved in recent years. According to the latest statistical bulletin of the Ministry of Transport of China, the total mileage of roadways in China has occupied first place globally, reaching 5.198 million kilometers. Asphalt pavement provides outstanding performance and ride comfort, accounting for more than 90% of the highways [1,2]. Nevertheless, as the mileage of asphalt pavements that must be maintained grows, the disease and disposal issues of asphalt pavement have drawn people’s attention. During the service life, the pavement will be impacted by loading, temperature, water, and other factors [3], causing the asphalt to undergo physical and chemical changes such as volatilization, oxidation, decomposition, and polymerization [4]. The performance of asphalt will deteriorate as a result of these consequences, the stress relaxation capability of the asphalt pavement declines as stiffness rises, and the asphalt pavement is susceptible to shrinkage fractures at low temperatures. The pavement quality will worsen if the cracks are not fixed or treated promptly, resulting in mesh cracks, cracks, rutting, and other more significant pavement diseases [5]. When the load interval or temperature rises, some microcracks will mend spontaneously (self-healing performance of asphalt), but other cracks that cannot be healed will spread due to vehicle load and form larger cracks. Water in the road environment, such as rainwater, will seep into the cracks and then gradually infiltrate into the bottom of the road under vehicle load, which will increase the water content of the road base and even the subgrade and diminish the strength.

There are several treatments for different kinds of cracks [6], such as slurry seal, microsurfacing, overlay, joint grouting, seam sticking, etc. Among these treatment approaches, joint grouting is the most extensively utilized due to its easy operation and high cost efficiency. At present, the asphalt pavement crack repair materials utilized in China generally comprise thermal asphalt, cold asphalt, and organic chemical specialist materials [7,8,9]. Polymer-modified asphalt, polymer-modified emulsified asphalt, and resin are widely employed as joint grouting materials in practical engineering applications [10]. However, from the standpoint of practical use, they have the faults of weak adhesion between grouting materials and fracture interface under water circumstances. To tackle this issue, a novel class of asphalt-based bionic material with good water resistance and strong adhesion was developed in this research.

Bionic design is an effective technique to emulate certain life traits and accomplish the intelligent design of materials. Mussels are a form of a marine creature with excellent adhesive capacity. The exceptional waterproof adhesion and universal adhesive qualities of mussels are particularly appealing. It is observed that mussels secrete various proteins via the Byssus, which are engaged in the adhesion process and are connected to the adhesion capacity and are solidified quickly after contact with saltwater to create an adhesive plate and securely bond to the substrate. The research discovered that the primary component of mucus is mussel adhesion protein (MAP) [11,12]. One of its most distinctive structural properties is the amino acid containing DOPA (dihydroxyphenylalanine). The catechol group (also known as catechol) in DOPA possesses chemical adaptability, versatility, and affinity diversity, which is the key to the super-strong adhesion property of mussels [13,14].

Due to the chemical versatility and affinity diversity of catechol functional groups, DOPA exhibits unique chemical properties. Thus, DOPA can boost the adhesion and cohesion of mussel adhesive proteins [15,16,17]. Through its chemical versatility, DOPA can undergo different reactions to achieve bonding and cross-link curing [18,19]. As shown in Figure 1, DOPA can be oxidized to Dopaquinone under certain conditions. After that, Dopaquinone can be rearranged and dehydrogenated to form Dehydro DOPA, which results in further cross-linking; DOPA quinone can also undergo Michael addition and Schiff base formation with amino and thiol groups; it can also form Carboxylated dihydroxyindole through intramolecular cyclization, and finally form cross-links. At the same time, Dopaquinone and DOPA can also be disproportionated to form free radicals, and finally coupled to tannins (Quinone tanning).

The magical adhesion properties of mussel adhesive proteins make them have unlimited potential applications in biomedical, national defense, marine engineering, and other fields. However, the ultra-low preparation quantity and expensive price limit the application of this excellent adhesive material. In order to obtain the adhesion ability equivalent to that of mussel adhesion protein in a wet state, people began to develop adhesion materials based on mussel adhesion protein. The study of adhesion materials using mussel adhesion protein was a gradual transition from natural mussel extraction to artificial synthesis. The first made use of the traditional protein extraction method and recombinant gene cloning method to prepare mussel adhesive materials. In the late 1980s, BioPolymers. Ltc successfully prepared mussel adhesive protein from mussel byssus glands by traditional extraction methods and developed a ‘Cell-Tak’ superadhesive [20,21]. ‘Cell-Tak’ has also been successfully used to enhance the adhesion of biomedical slides, plastics, or metal surfaces to cellular tissue samples. Subsequently, Hwang et al. recombined, expressed and purified the genes of Mgfp-5 and Mgfp-3A mussel adhesion proteins by recombination technology to obtain a superadhesive like ‘Cell-Tak’ [22,23]. The recombinant proteins with adhesion properties similar to those of ‘Cell-Tak’ were obtained. However, the cost of recombinant gene technology is also surprising due to the fact that the regulatory mechanism is still unclear. Lv et al. also tried to perform some work on the extraction of natural protein and genetic engineering preparation of mussel adhesive materials [24]. Although mussel adhesion protein can be obtained by direct extraction or gene recombination, the separation and purification of mussel adhesion protein from mussels is an extremely complex process with a meager yield. This makes it very challenging to directly apply this adhesive to commercial applications, however, it also further highlights the urgency of developing synthetic mussel bionic adhesives with strong waterproof bonding properties. According to the intrinsic adhesion mechanism of mussel adhesive protein (DOPA, the key factor of adhesion performance), this paper aims to prepare polymeric adhesives to mimic the effect of natural mussel adhesion proteins.

Although biomimetic materials have been used in asphalt materials, this work marks the first time the bionic mussel adhesion protein has been applied to asphalt materials. By analyzing its impact on the overall performance of asphalt, this study aims to investigate the viability of using bionic mussel adhesion protein in asphalt. L-Dopa Methacrylic anhydride (L-DMA) was synthesized based on the features of bionic mussel adhesion protein in this study and used to modify 70# base asphalt with varied quantities to prepare L-DMA-modified asphalt (L-DMA-A). The thermal stability and structural composition of L-DMA were investigated. In addition, the high-temperature rutting resistance and low-temperature cracking resistance of L-DMA-modified asphalt were examined by DSR and BBR. Finally, the modification mechanism of L-DMA-modified asphalt was analyzed by FTIR. The flowchart of the experiment design is shown in Figure 2.

## 2. Materials and Experiments

### 2.1. Raw Materials

In this study, the chemicals used to prepare L-DMA included sodium tetraborate (Na_2_B_4_O_7_•10H_2_O), sodium bicarbonate (NaHCO_3_), Levodopa (L-DOPA), methacrylic anhydride (MAA), magnesium sulfate (MgSO_4_), ethyl acetate (EAc), tetrahydrofuran (THF), and cyclohexane. The first four chemicals were purchased from Aladdin Ltd. (Shanghai Aladdin Biochemical Technology Co. Ltd., Shanghai, China) and the latter three were purchased from Sinopharm Chemical Reagent Co. Ltd., Shanghai, China. All chemicals were chemically pure and used without further purification. The asphalt used, 70# base asphalt, was from Hubei Guochuang Hi-tech Materials Co. Ltd., of China (Wuhan), and the basic properties are shown in Table 1.

### 2.2. Synthesis of L-DMA

The mechanism of the synthesis process of L-DMA [25] is shown in Figure 3. An amount of 24 g of Na_2_B_4_O_7_•10H_2_O and 10 g of NaHCO_3_ were accurately weighed and dissolved in 200 mL deionized water and aerated with Ar for 20 min, then 10.4 g of L-DOPA (52.8 mmol) was added and continued to be bubbled with Ar. After 5 min, 9.5 g of MAA (58.1 mmol) was dissolved in 50 mL of THF and slowly dropped to the above solution. During the dropwise addition, 1 mol/L NaOH solution was used to keep the pH of the solution above 8. The reaction mixture continued to be stirred for 18 h at room temperature and protected from light.

The reaction mixture was filtered by extraction to remove the precipitate from the bottom of the flask, and then the pH was adjusted to less than 2 with 4 mol/L HCl solution, followed by extraction with 100 mL EAc. The organic phase was collected after three extractions with 100 mL of EAc, and the aqueous phase was removed. An amount of 4 g of anhydrous magnesium sulfate was added to the organic phase, stirred for 12 h, and filtered to remove the residual water. Finally, the organic phase was concentrated by a rotary evaporator to 50 mL.

Finally, the concentrated ethyl acetate solution was slowly dropped into 400 mL of cyclohexane with high-speed stirring (at 2000 rpm). After the dropwise addition, the solution was stirred for 30 min, and then placed at 4 °C for 18 h to allow complete recrystallization. Finally, the solid was filtered out and dried in a vacuum oven. The synthesis process is shown in Figure 4.

### 2.3. Chemical Structure and Thermal Stability Characterizations of L-DMA

FTIR spectra of MAA, L-DOPA, and L-DMA were recorded on a Thermo Nicolet 6700 Fourier-transform spectrophotometer (Waltham, MA, USA). In all cases, 16 scans were collected at a resolution of 2 cm^−1^ for the spectrum in the range of 4000–400 cm^−1^.

TGA was used to determine the thermal stability of L-DMA, which was carried out using the synchronous thermal analyzer (TA TGA55, New Castle, DE, USA) The L-DMA was protected in an N_2_ atmosphere at a flow rate of 20 mL/min, in the temperature range from room temperature to 800 °C at a heating rate of 10 °C/min.

### 2.4. Preparation of L-DMA-Modified Asphalt

We added a certain mass ratio of L-DMA into the heated 70# base asphalt and mixed manually for ten minutes. To guarantee a good dispersion of L-DMA, a shear apparatus was used to disperse L-DMA at a rate of 4000 r/min for 40 min at a temperature of 170 °C. The dosage of L-DMA was 1%, 2.5%, 5%, 7.5%, and 10% of asphalt, respectively.

### 2.5. Property Characterizations of L-DMA-Modified Asphalt

#### 2.5.1. DSR Test

DSR (MCR-101, Anton Paar, Graz, Austria) examined the rheological properties of L-DMA-A. Asphalt’s viscoelasticity is significantly impacted by exposure temperature changes as a typical viscoelastic material [26,27]. Specific test parameter settings for the temperature sweep are shown in the Table 2.

Asphalt pavement will be affected by vehicle load during the service process. Loading with different frequencies will cause different deformations of asphalt pavement, and the deformation size determines the high-temperature performance of asphalt pavement. Viscosity is an essential parameter of asphalt and the addition of L-DMA will affect the viscosity of asphalt binder. Therefore, it is necessary to study the variation trend of the viscosity of 70# asphalt and L-DMA-modified asphalt under different load frequencies. The specimens were subjected to a frequency sweep in the range of 0.01–10 Hz. In order to meet the practical application of pavement, the temperature was set to 58 °C.

Additionally, L-DMA-modified asphalt with different L-DMA contents were tested by loading–unloading mode in the Multiple Stress Creep and Recovery (MSCR) test, which can be used to characterize the high-temperature performance of asphalt [28]. In this paper, the MSCR tests under stresses of 0.1 kPa and 3.2 kPa were carried out at 58 °C. The whole test contained 20 cycles of creep and recovery. Specifically, the first 10 cycles were conducted under the stress of 0.1 kPa, and the following 10-cycle test under the constant stress of 3.2 kPa was further conducted to finish the whole test. Figure 5 shows the creep and recovery (loading–unloading) curves of asphalt in one cycle.

According to AASHTO M 332 Standard [29], the two evaluation indexes obtained by the MSCR test are the creep recovery rate *R* (%) and nonrecoverable creep compliance $J_{nr}$, which are used to evaluate the viscoelastic characteristics of asphalt under high-temperature conditions. The nonrecoverable creep compliance $J_{nr}$ is the ratio of ultimate unrecoverable strain and applied stress in each loading cycle, which describes the contribution of asphalt to high-temperature deformation of asphalt mixtures. The creep recovery rate R (%) is the percentage of recoverable strain and total creep strain in the loading cycle, which characterizes the high-temperature elastic recovery performance of asphalt and reflects the influence of delayed elastic deformation of asphalt to a certain extent. According to the Equations (1)–(3) below, *R* (%) and $J_{nr}$ under different stresses can be calculated.
(1)$$\begin{array}{c} J_{nr100}\left(\frac{1}{kPa}\right)=\frac{1}{10}\left\{\sum_{1}^{10}\frac{non-recoverable\;strain}{0.1}\right\} \end{array}$$
(2)$$J_{nr3200}\left(\frac{1}{kPa}\right)=\frac{1}{10}\left\{\overset{10}{\underset{1}{\sum }}\frac{non-recoverable\;strain}{3.2}\right\}$$
(3)$$R\left(\%\right)=\frac{1}{10}\left\{\overset{10}{\underset{1}{\sum }}\frac{recoverable\;strain}{peakstrain}\times 100\%\right\}$$

#### 2.5.2. BBR Test

Low-temperature cracks are inevitable in the use of asphalt pavement. Improving the low-temperature performance of asphalt pavement is the top priority of road workers. In general, BBR is most commonly used for the low-temperature performance evaluation of asphalt. According to the specification AASHTO T313-12 [30], 70# asphalt and L-DMA-modified asphalt beams were prepared, respectively, then held 1 Dh at the test temperature and were subjected to a 4 min loading test to measure the creep stiffness modulus S and creep rate m. S and m were selected at 60 s, if S ≤ 300 MPa and m ≥ 0.3 at this condition, then the asphalt is applicable at this temperature condition.

#### 2.5.3. FTIR Spectroscopy Analysis

To study the chemical structure changes of L-DMA after modified asphalt and the modification mechanism, the FTIR Spectroscopy test was carried out. The sample was dissolved in carbon disulfide and then dropped onto a clear KBr lens. After drying for a moment, the sample was prepared and put into the instrument for testing.

## 3. Results and Discussion

### 3.1. Structural Analysis and Thermal Stability Analysis of L-DMA

The infrared spectra of L-DMA are shown in Figure 6. The absorption peak at 3370 cm^−1^ was the vibration peak of catechol phenolic hydroxyl, and the formation of hydrogen bonds widened the peak area. The N-H stretching vibration peak appeared at 3202 cm^−1^. The C=O stretching vibration peak at 1730 cm^−1^ originated from the amide group, and the hydrogen bond made the electron shift outward and the bond force weaken. The bending vibration and deformation vibration of sec-amide appeared at 1655 cm^−1^ and 1520 cm^−1^, respectively. However, the stretching vibration of C=C overlapped with the bending vibration of N-H, so the absorption peak appeared at 1655 cm^−1^. At the same time, the absorption peaks at 1605 cm^−1^ and 1450 cm^−1^ were also observed, which were the stretching vibration of the L-DOPA aromatic ring. The single peak at 1260 cm^−1^ is the C-N stretching vibration.

Compared with the spectra of the reaction monomers MAA and DOPA, in the FTIR spectra of L-DMA, the absorption peaks of methacrylic anhydride in MAA at 1787 and 1725 cm^−1^ had obviously disappeared and changed into a single peak of C=O at 1730 cm^−1^, and two amino peaks at 3210 and 3075 cm^−1^ at L-DOPA became one, indicating that the primary amine in L-DOPA reacted to form the secondary amide. From the above analysis, MAA and L-DOPA reacted successfully to form L-DMA.

Figure 7 shows the thermal weight loss curve of the L-DMA monomer powder, the initial thermal decomposition temperature of L-DMA is 221.5 °C with a mass loss of 5%, which is mainly due to the volatilization of bound water. There was a sharp mass loss between 224 °C and 478.8 °C, which corresponded to the decomposition of the C-N bond and methacrylate. Subsequently, the slow mass loss of the sample from 480 °C to 800 °C was the decomposition of the L-DMA main structure. The maximum temperature during asphalt mixing and paving is about 180 °C. Therefore, L-DMA can be used for asphalt modification.

### 3.2. Modification Mechanism of L-DMA

The modification mechanism of L-DMA-modified asphalt was analyzed by FTIR. Figure 8 compared the infrared spectra of L-DMA, 70# asphalt, and L-DMA-modified asphalt. It can be found that the characteristic absorption peaks of 70# base asphalt show asymmetric and symmetric stretching vibration of methylene at 2925 and 2854 cm^−1^. C=C and C=O stretching vibrations appear at 1600 cm^−1^; the absorption peaks of the benzene ring stretching vibration are 866 and 809 cm^−1^. Combined with the analysis of the infrared spectrum of L-DMA in Figure 6, it can be inferred that the modification process of L-DMA on asphalt is mainly a physical modification process. With the addition of L-DMA, the conjugated effect between catechol and the aromatic benzene ring in asphalt resulted in the redshift of the vibration absorption peak of the benzene ring. The peak area widened at about 3370 cm^−1^, suggesting that L-DMA can generate many hydrogen bonds with free hydroxyl and amino groups in asphalt through catechol groups, thus forming a strong hydrogen bond and promoting its combination with asphalt. The number of hydrogen bonds increased significantly with an increase in L-DMA content. The hydroxyl characteristic peak area of modified asphalt with 10% L-DMA content was larger than that of other contents. Moreover, due to the existence of the benzene ring, L-DMA can interact with the aromatic ring in asphalt through the Π-Π accumulation [31,32,33]. These can improve the adhesion performance of modified asphalt. At the same time, dense hydrogen bonds can also enhance the self-healing performance of asphalt [34,35,36,37]. Infrared spectra also show that L-DMA does not decompose and exists stably in asphalt.

### 3.3. Rheological Properties of L-DMA-Modified Asphalt

When L-DMA is added to asphalt as a modifier, physical and chemical modifications will inevitably occur, thus affecting the original structure and the viscoelastic properties of asphalt. In general, the composite shear modulus G* is the measurement of the total resistance of the material during repeated shear deformation [38,39]; the phase angle δ is the relative index of the number of recoverable and irreversible deformations.

The composite modulus and phase angle curves of L-DMA-modified asphalts with different contents of L-DMA in the temperature range of 30–80 °C are shown in Figure 9. It is clear that with an increase in L-DMA content, the composite modulus G* of modified asphalt increases and the phase angle δ decreases, indicating that the addition of L-DMA improves the shear deformation resistance of asphalt, and the elasticity of asphalt is obviously enhanced. The modified asphalt with 10% content has the most excellent high-temperature performance, which is mainly caused by the good adhesion role of L-DMA in asphalt. The catechol group in L-DMA generated a large number of hydrogen bonds with the free hydroxyl and amino groups in asphalt, thus forming a strong hydrogen bond and promoting the integration with asphalt [40,41]. Moreover, catechol and the asphalt aromatic ring accumulated to produce cohesion [42,43], increasing the proportion of elastic components in the modified asphalt and improving the shear deformation resistance of asphalt.

The evaluation index of anti-rutting performance of asphalt is quantified by the rutting factor, which is expressed as G*/sinδ. In general, the asphalt binder with a larger rutting factor shows stronger anti-rutting ability [44]. According to Superpave specification, AASHTO: MP1, at the highest pavement design temperature, the G*/sinδ value of matrix asphalt should be at least 1.0 kPa [45]. Figure 10 shows the rutting factors of base and different modified asphalts at the temperature range of 30–80 °C. The results show that the rutting factors of modified asphalts rise with an increase in L-DMA content. The modified asphalt with 10% content has the largest rutting factor because the incorporation of L-DMA will introduce phenolic hydroxyl groups, which will form hydrogen bonds with the aromatic components in the light component of the asphalt, thereby increasing the adhesion of the material. At high temperatures, the catechol group in L-DMA will be oxidized into semiquinone and quinone structures, which improves the overall cohesive energy [46,47]. In addition, the benzene ring can also produce Π-Π conjugate with the aromatic ring in asphalt, which increases the cohesive energy. These effects increase the cohesion and adhesion of modified asphalt. These enhanced bonding structures will hinder the free flow of modified asphalt, enhance the elastic recovery ability, and improve the rutting resistance.

The low-temperature rheological property of asphalt binder is one of the important factors for its cracking resistance. The SHRP standard defines G*sinδ as the evaluation index of fatigue cracking. The smaller the value of G*sinδ, the better the ability to resist fatigue damage [48,49,50]. In this paper, the low-temperature rheological properties of 70# base asphalt and modified asphalt with different contents of L-DMA were tested in the temperature range of −10–30 °C and the test results are shown in Figure 11. The composite modulus of modified asphalt decreases with an increase in temperature, while the phase angle increases. When the content of L-DMA is 1%, the modified asphalt has the highest composite modulus. Then, with an increase in L-DMA content, the composite modulus of modified asphalt shows a downward trend. When the content of L-DMA is 10%, the composite modulus of modified asphalt is the lowest and is lower than that of 70# asphalt. The reason is the change of L-DMA’s effect, and when the content is low, it mainly plays a role of filling in asphalt, which makes the low-temperature performance of modified asphalt worse. However, with an increase in L-DMA content, unsaturated rings such as benzene, quinone, and pyridine in aromatic components of asphalt will conjugate with the benzene ring in L-DMA, and the rings overlap with each other to form a layered structure, which reduces the composite modulus G* of modified asphalt.

Figure 12 shows the changing trend of the fatigue factors of 70# asphalt and modified asphalts with different contents of L-DMA at different temperatures. The results show that the fatigue factor of modified asphalt increases first and then decreases with an increase in L-DMA content. At a high content of 10%, the addition of L-DMA decreases the fatigue factor and enhances the fatigue resistance of asphalt due to the conjugation between the unsaturated rings of L-DMA and asphalt.

The viscosities of 70# asphalt and L-DMA-modified asphalts mixed with different contents of L-DMA are shown in the Figure 13. It can be seen from the figure that with an increase in loading frequency, the viscosity of asphalt at 58 °C shows a downward trend. The viscosity of 70# asphalt is the lowest, while 10% L-DMA-modified asphalt has the highest viscosity. The viscosity can reflect the deformation resistance of the material to a certain extent, and the viscosity of 10% L-DMA-modified asphalt can still maintain a high level at 58 °C, which indicates that the addition of L-DMA contributes greatly to the improvement of the high-temperature performance of asphalt.

The strain changes of 70# base asphalt and L-DMA-modified asphalt under different stress levels at 58 °C are shown in Figure 14. The result shows that the 70# base asphalt has the largest cumulative strain. With an increase in L-DMA content, the cumulative strain of the modified asphalt gradually decreases. This is because the cohesion between L-DMA and asphalt improves the elasticity of asphalt materials, significantly improving the recoverable deformation ability of asphalt and thus improving the high-temperature performance of asphalt.

Table 3 shows the experimental results of creep recovery rate *R* (%) and unrecoverable creep compliance ($J_{nr}$) of base asphalt and modified asphalts at different stress levels. The results show that under the action of 100 Pa and 3200 Pa, the creep recovery rate R (%) of L-DMA-modified asphalt increases with an increase in L-DMA content, indicating that the addition of L-DMA enhances the recovery deformation ability of asphalt. The unrecoverable creep compliance ($J_{nr}$) decreases with an increase in L-DMA content, indicating that the incorporation of L-DMA improves the permanent deformation resistance of asphalt, which is consistent with the results of the rutting factor test. Comparing the unrecoverable creep compliance ($J_{nr}$) of asphalt and L-DMA-modified asphalts under different stress levels, it can be found that the unrecoverable creep compliance $\left(J_{nr}\right)$ of 70# base asphalt changes greatly under 100 Pa and 3200 Pa, while the unrecoverable creep compliance $\left(J_{nr}\right)$ of L-DMA-modified asphalts does not change significantly under different stress levels. It indicates that the addition of L-DMA reduces the dependence of asphalt on shear stress and enhances the shear deformation resistance of asphalt. The hydrogen bonds generated by the addition of L-DMA have certain reversibility, so the creep recovery rate *R* (%) of L-DMA-modified asphalt has greatly improved.

### 3.4. Low-Temperature Crack Resistance of L-DMA Modified Asphalt

It can be seen from Figure 15 that with a decrease in temperature, the S modulus increases but the m value decreases, indicating that the lower the temperature, the worse the low-temperature performance of asphalt. When the temperature is lower than −12 °C, the m of 70# asphalt and L-DMA-modified asphalt will be far less than 0.3, and the S is much higher than 300 MPa, indicating that when the temperature is lower than −12 °C, the low-temperature performance of 70# asphalt and L-DMA-modified asphalt cannot meet the standard. At −12 °C, the S and m of 70# asphalt basically meet the requirements of the specification. It can be seen that with an increase in L-DMA content, the S modulus increases first and then decreases gradually, indicating that less L-DMA content will affect the deformation capacity of asphalt. However, with an increase in L-DMA content, the deformation capacity of modified asphalt at low temperatures is becoming better and better, and the deformation capacity of modified asphalt is the best when the content of L-DMA is 10%, meeting the standard. The m decreases first and then increases, suggesting that with the incorporation of L-DMA, the stress relaxation ability of asphalt to external load decreases first and then increases. When the content of L-DMA is 10%, the m is the largest, and the modified asphalt has the best stress relaxation ability to deal with external loads.

Combined with the low-temperature rheological test results of DSR, it is found that the low content of L-DMA has an adverse effect on the low-temperature performance of asphalt, but with an increase in L-DMA content, it has a significant improvement on the low-temperature performance of asphalt. This is mainly because the low content of L-DMA plays a filling role in asphalt, increasing the composite modulus G* and creep stiffness modulus S of L-DMA-modified asphalt, reducing the creep rate m, and worsening the low-temperature performance of modified asphalt. However, with an increase in L-DMA content, the unsaturated rings such as benzene, quinone, and pyridine in the aromatic components of asphalt will conjugate with the benzene ring in L-DMA, and the rings overlap with each other to form a layered structure. The interlayer friction is small, which reduces the G* value of modified asphalt, decreases the S value, and increases the m value, thereby improving the low-temperature performance of asphalt. This is consistent with the results of the low-temperature sweep test in DSR.

## 4. Conclusions

In this study, the mussel adhesion protein material L-DMA was synthesized and used to modify asphalt. The deformation and cracking resistances of L-DMA-modified asphalt were studied by DSR temperature sweep, frequency sweep, multi-stress creep tests, and BBR tests. In addition, the modification mechanism of L-DMA-modified asphalt was analyzed by FTIR test. According to the test results, the following conclusions can be drawn.

Comparing with MAA and L-DOPA, the infrared spectrum of L-DMA has new characteristic peaks at 3370 cm^−1^, 3202 cm^−1^, 1730 cm^−1^, 1655 cm^−1^, 1520 cm^−1^, 1605 cm^−1^, 1450 cm^−1^, and 1260 cm^−1^, which proves that L-DMA was synthesized successfully. Moreover, L-DMA has good thermal stability and can be used in asphalt materials.The modification of L-DMA to asphalt is mainly a physical process. Hydrogen bond and aromatic ring conjugation introduced by catechol can improve the adhesion of asphalt.L-DMA has a noticeable improvement impact on the high-temperature performance of asphalt due to the strong hydrogen bonding, conjugation of benzene rings, and transformation of quinone and semiquinone by catechol at high temperatures. Compared with 70# asphalt, L-DMA-modified asphalt has a higher composite modulus, rutting factor, viscosity, and creep recovery rate.The impact of LDMA on the low-temperature fracture resistance of asphalt is complicated. With the rise of L-DMA content, the low-temperature performance of modified asphalt drops initially and later increases. When the L-DMA content reaches 10%, the low-temperature fracture resistance of L-DMA modified asphalt will be higher than that of 70# asphalt due to the layered structure formed by aromatic ring conjugation.

Overall, the addition of L-DMA has a positive impact on the high-temperature rutting resistance and a high content of L-DMA will have some improvement on the low-temperature cracking resistance of asphalt. The usage of L-DMA as a kind of mussel adhesion protein material improves the performance index of asphalt, and, theoretically, there is a possibility to improve water resistance and self-healing properties of asphalt, which will be explored in future research. Furthermore, due to the high cost and complex preparation process of L-DMA, further research is needed to decrease the cost and increase the preparation efficiency.

## Figures and Tables

**Figure 1 materials-15-05351-f001:**
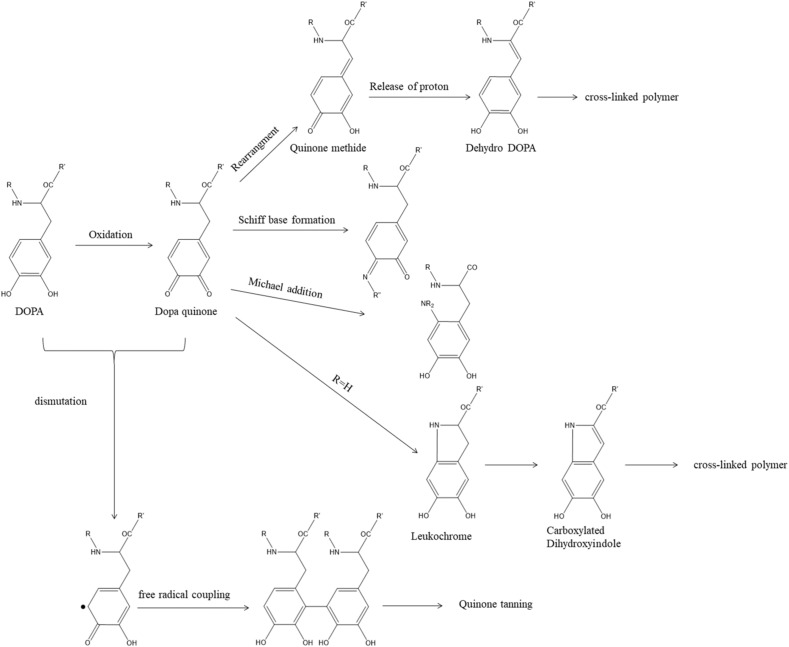
Possible cross-linking pathways of DOPA.

**Figure 2 materials-15-05351-f002:**
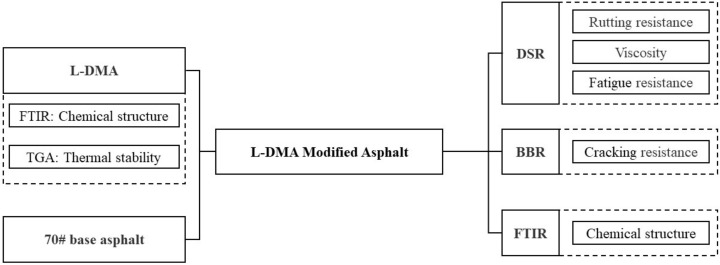
Flowchart of the experiment design.

**Figure 3 materials-15-05351-f003:**
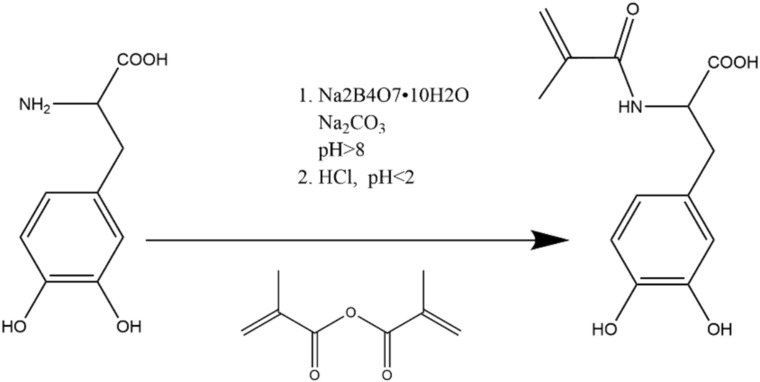
Mechanism of the synthesis of L-DMA.

**Figure 4 materials-15-05351-f004:**
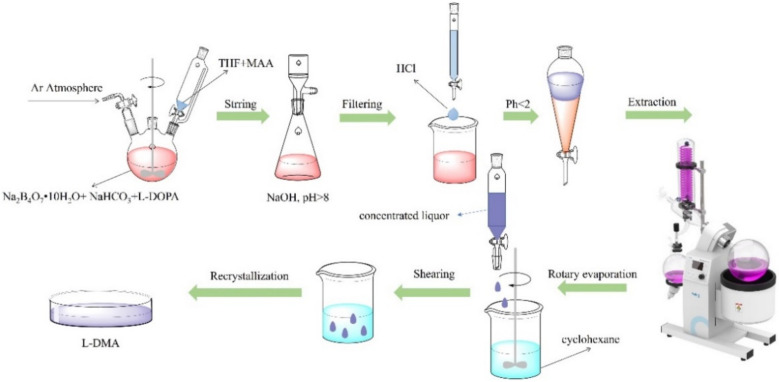
Preparation procedure of L-DMA.

**Figure 5 materials-15-05351-f005:**
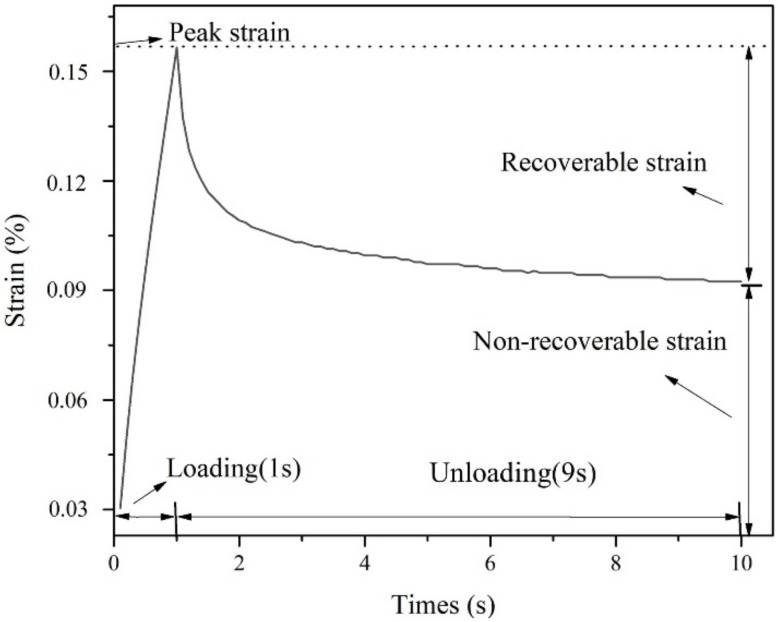
Change of strain with time during one creep—recovery cycle.

**Figure 6 materials-15-05351-f006:**
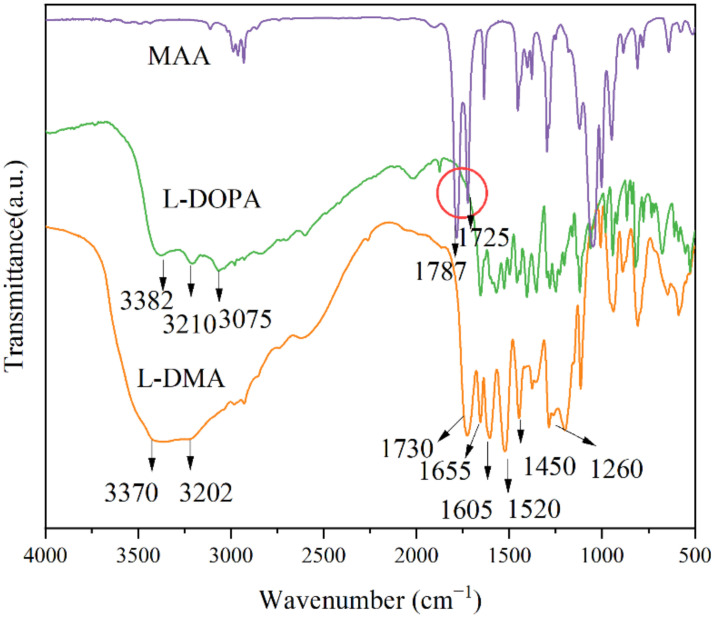
FT-IR spectra results of MAA, L-DOPA, and L-DMA.

**Figure 7 materials-15-05351-f007:**
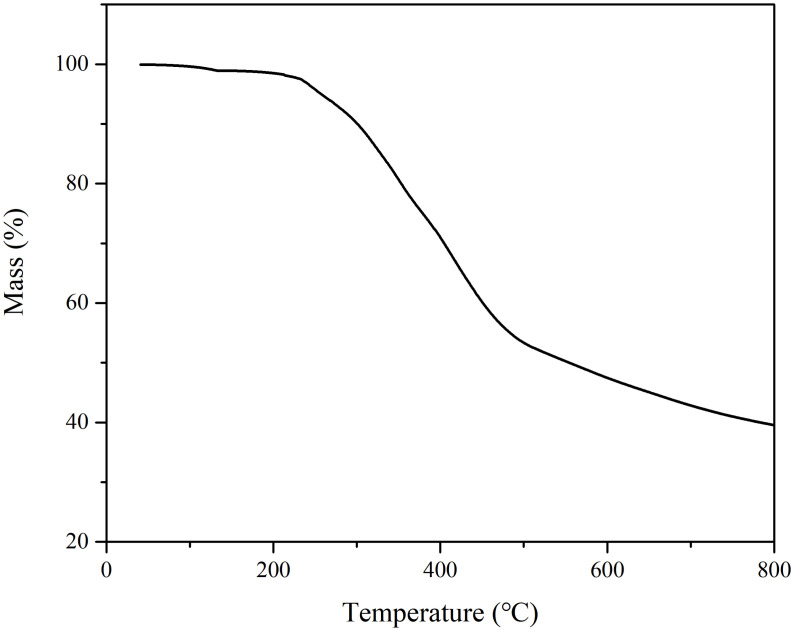
The TG curve of L-DMA.

**Figure 8 materials-15-05351-f008:**
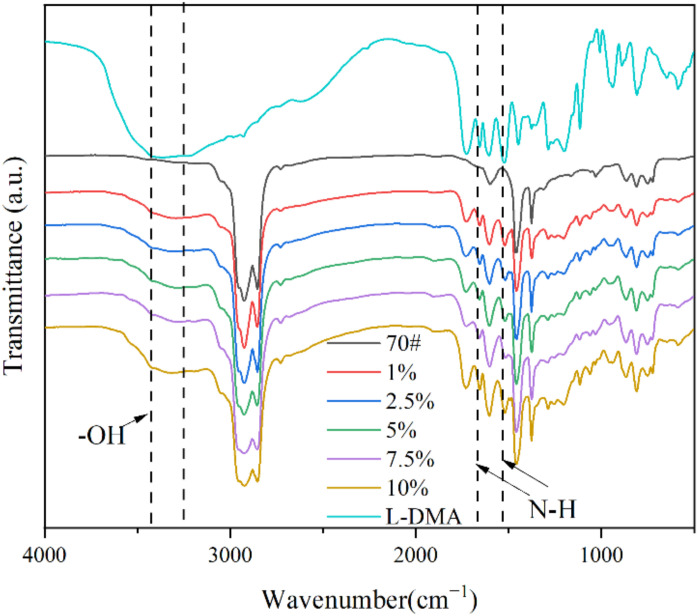
FT-IR spectra results of L-DMA, 70# base asphalt, and L-DMA-A.

**Figure 9 materials-15-05351-f009:**
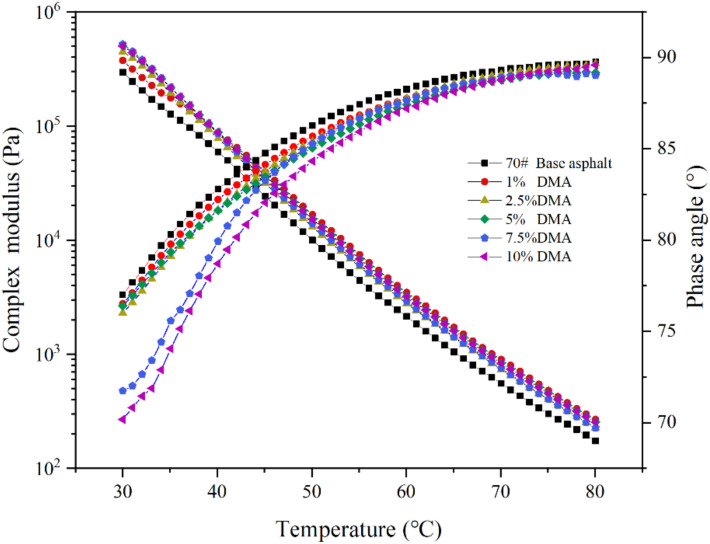
The result of the temperature sweep test (30–80 °C).

**Figure 10 materials-15-05351-f010:**
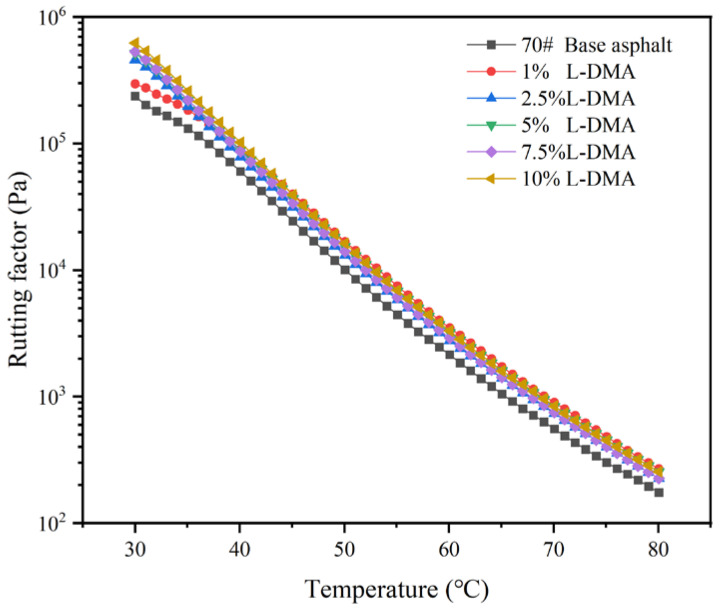
Rutting factor and temperature curves of different asphalts.

**Figure 11 materials-15-05351-f011:**
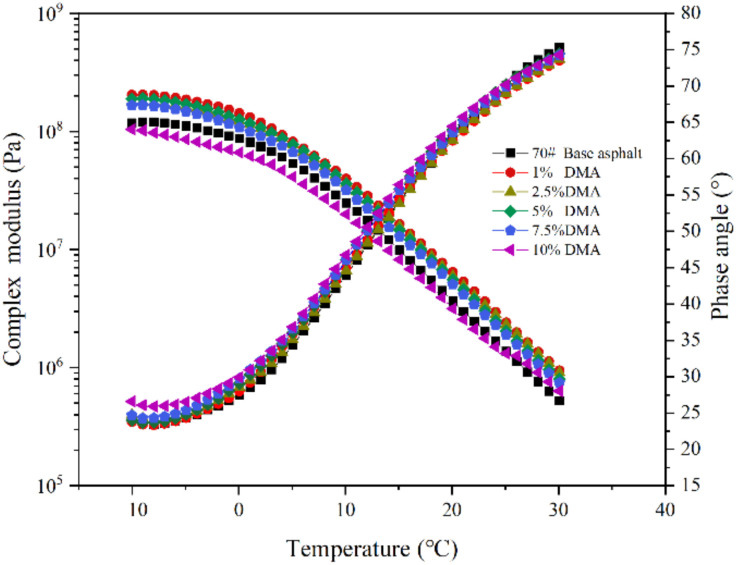
The result of the temperature sweep test (−10–30 °C).

**Figure 12 materials-15-05351-f012:**
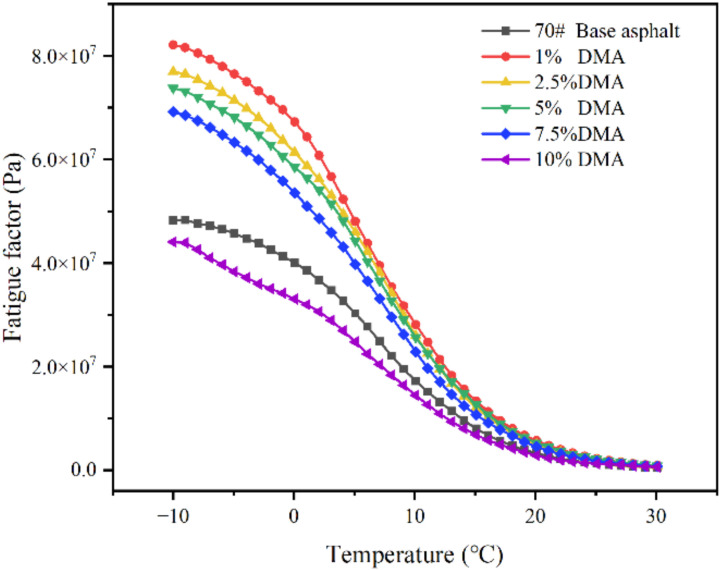
Fatigue factor and temperature curves of different asphalts.

**Figure 13 materials-15-05351-f013:**
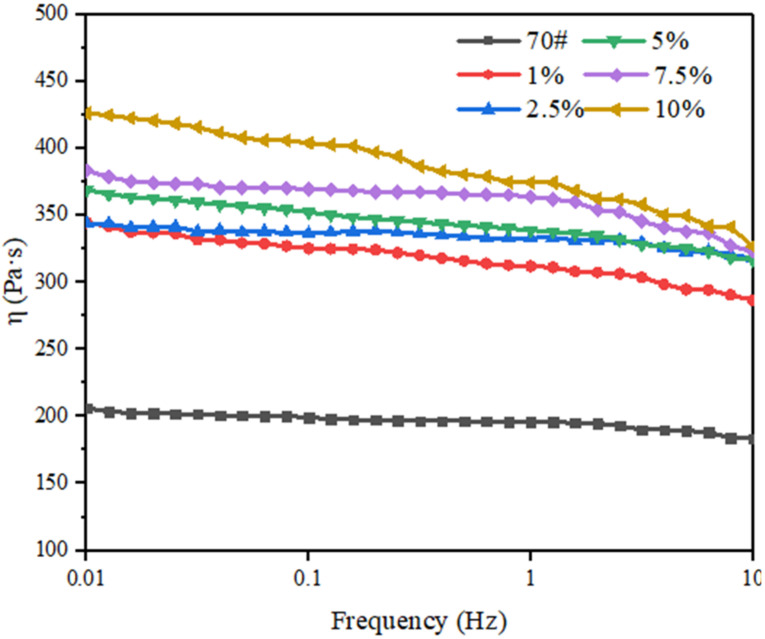
Frequency—Viscosity Curve at 58 °C.

**Figure 14 materials-15-05351-f014:**
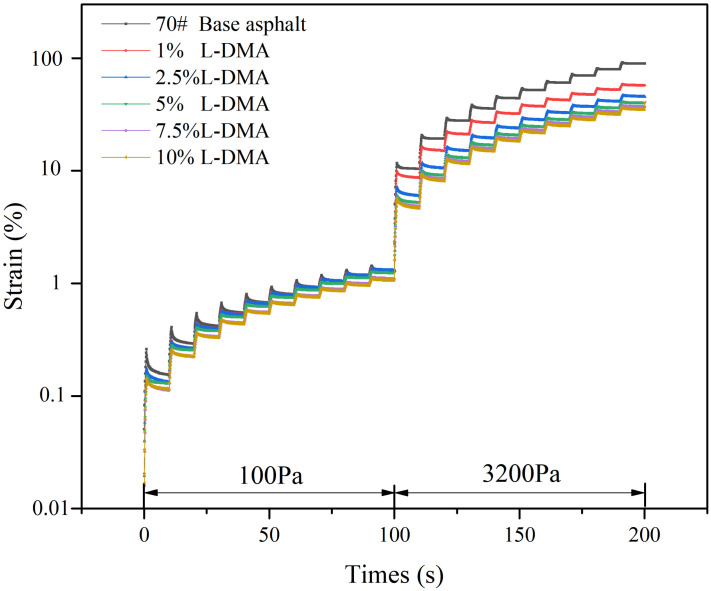
Strain during the MSCR test at 58 °C.

**Figure 15 materials-15-05351-f015:**
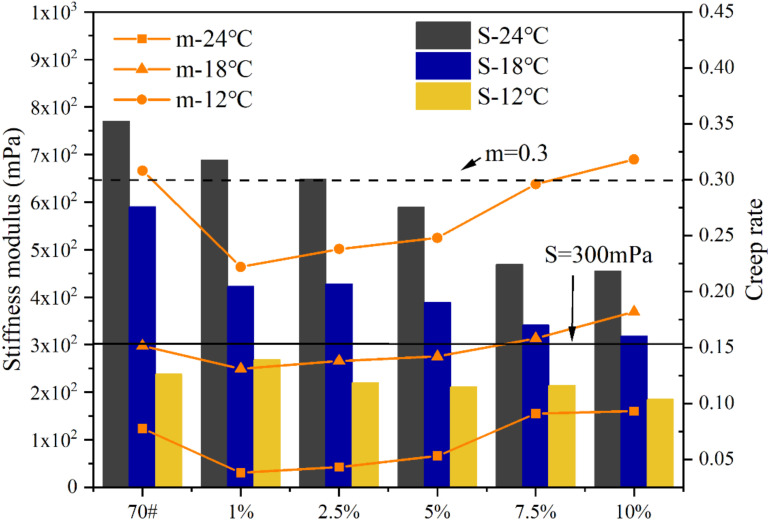
The result of BBR.

**Table 1 materials-15-05351-t001:** Properties of the 70# base asphalt.

Properties	Test Value	Requirements
Penetration (25 °C, 0.1 mm)	68.9	60–80
Ductility (15 °C, cm)	184	≥100
Softening point (°C)	48.4	≥46
Density (g/cm^3^)	1.034	-

**Table 2 materials-15-05351-t002:** Test parameters of temperature sweep.

Temperature	Plate Diameter	Plate Gap	Shear Frequency	Heating Rate
−10–30 °C	25 mm	1 mm	10 rad/s	2 °C/min
30–80 °C	8 mm	2 mm		

**Table 3 materials-15-05351-t003:** The MSCR parameters for asphalt samples.

Samples	*R*-100	*R*-3200	$J_{nr}$-100	$J_{nr}$-3200
70# Base asphalt	4.05	2.55	2.66	3.56
1% L-DMA	5.67	2.95	2.62	2.75
2.5% L-DMA	6.29	3.58	2.48	2.43
5% L-DMA	6.32	3.62	2.18	2.23
7.5% L-DMA	6.49	3.91	2.14	2.17
10% L-DMA	7.32	4.11	1.78	1.83

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
