# Peer review of "Bionic Synthesis of Mussel-like Adhesive L-DMA and Its Effects on Asphalt Properties"

_materials, 2022, doi:10.3390/ma15155351_

Round 1
Reviewer 1 Report
The topic of the presented study is quite relevant, since it deals with issues related to increasing the performance properties of an asphalt pavement using L-DMA mead-like adhesive. The paper investigated the rheological properties and low-temperature properties of asphalt modified with L-DMA. Also, the modification mechanism was investigated using FTIR. As a result, the study showed that L-DMA improves the properties of asphalt and can also potentially improve the water resistance of asphalt.
However, it should be noted that the introduction does not address other ways to improve the performance of asphalt and related building materials. In this regard, I recommend considering the following works in the literature review (1. Kuz'min M.P., Kuz'mina M.Yu., Kuz'mina A.S. Production and properties of aluminum-based composites modified with carbon nanotubes // Materials Today: Proceedings. - 2019. - V. 19 (5), - P. 1826–1830. 2. Kuz'min M.P., Larionov L.M., Kondratiev V.V., Kuz'mina M.Yu., Grigoriev V.G., Kuz'mina A.S. Use of the burnt 3. Kuz'min M.P., Larionov L.M., Kondratiev V.V., Kuz'mina M., rock of coal deposits slag heaps in the concrete products manufacturing, Construction and Building Materials, 2018. Yu., Grigoriev V.G., Kuz'mina A.S. Burnt rock of the coal deposits in the concrete products manufacturing // Magazine of Civil Engineering - No. 08 (76) - P. 170–181).
Reviewer 2 Report
l To avoid misunderstandings, the acronym should be explained when it first appears, e.g., FTIR, TGA in the abstract. (Page 1)
l For the convenience of readers and completeness of experimental introduce, the test method of cited standard specification is better to introduce, and the provenances of the standard specification also should be written in the references. E.g., AASHTO T313 – 12, AASHTO MP 19-10. (Page 8,9)
l In the first paragraph of page 16, “The results reply that the fatigue performance of modified asphalt decreases first and then rises with the increase of L-DMA content” what is the basis of the fatigue performance rise trend?
l Figure 11 shows fatigue factor curves of different asphalts form -10-30℃, how about the fatigue factor curves in 30-80℃?
l This manuscript introduced the bionic mussel-like adhesive modification asphalt properties. Did the author have research for the modification asphalt properties that directly use mussel adhesion protein (MAP)? Is it the same or better than the bionic mussel-like adhesive modification asphalt?
Reviewer 3 Report
Please find my comments below:
1. I think it is more appropriate to write L-DMA in a full name
2. Please add the reason why there is no purification process for all the chemicals used in the study.
3. Please correct FT-IR in Section 2.3
4. Please improve the paper formatting. Some of the sentences are using different types of fonts and some of the sentences are in double spacing and single spacing. Please ensure the formatting is consistent.
5. Please use appropriate terminology, Asphalt Mortar, in Page 7. Please change mortar to other appropriate words.
6. Please add signposting after 3.2 section
7. How do you ensure the use of L-DMA will be working if it is applied in real road construction. The results show encouraging results in the lab. However, moisture, weather, traffic loading and etc factor will affect the performance of this material as a crack sealing material.
8. What are the limitations of this material?
9. Obvious self-citation from one of the co-authors shown in the references list.
Reviewer 4 Report
The article titled “Bionic synthesis of mussel-like adhesive L-DMA and its effects on asphalt properties” is an interesting approach to the cracking problem in asphalt pavements using innovative L-DMA. The abstract section is written clearly. The introduction section provides a comprehensive literature review, novelty, and research significance. Materials and methods described the all materials and experimental testing adequately. All necessary properties of materials and preparation method of L-DMA is mentioned. The description of results and discussion is good. All sections are written clearly and smooth flow of information. I think that article has merit for publication.
Reviewer 5 Report
General comments: The paper entitled ‘Bionic synthesis of mussel-like adhesive L-DMA and its effects on asphalt properties’ addresses issues related to asphalt grouting under wet conditions using bionics approaches to reproduce the adhesion capacity of mussels. The paper is recommended for publication after minor revision. Please address the specific comments outlined as follows:
Specific comments:
· Punctuation and symbols: Please, review punctuations and symbols throughout the manuscript. There are inconsistencies throughout the manuscript potentially.
· Citations and references: Please, double-check all the citations and cross-references throughout the manuscript. References like ‘Lv[24]’ must be revised to include the name of the authors.
· Aim: The aim of the study is stated correctly at the end of the first paragraph of the Introduction section; however, the paragraph must be revised by stating the knowledge gap followed by the aim, rather than saying what was done in the study. The details of what the study encompasses will be described in the subsequent sections.
· Materials and Experiments: The Materials and Experiments sections must include an initial subsection describing the method overview used to achieve the aim of the study. Please, include a graphical summary (flow chart) for this purpose.
· Cost: As described in the introduction, grouting techniques are usually used to treat asphalt cracking due to their cost efficiency. Considering the large scale of roads and the need for cost-efficient solutions, the authors should provide information about the expected cost of this new technology when applied at scale. Please, include a section to discuss the implications of the developed technol.
Round 2
Reviewer 2 Report
1. In figure 6, the red circle showed the peak disappeared at 1787cm-1 and 1725cm-1, but in line 228 “the absorption peaks of methacrylic anhydride in MAA at 1787 and 1752 cm-1 were...”. Does author mean the absorption peaks of MMA disappear at 1787cm-1 and 1752cm-1 not 1787cm-1 and 1725cm-1?
2. There are still some grammar mistakes. For the fluency of the article, it’s better to do a grammar check.
Reviewer 3 Report
Thank you for amending the paper based on the comments. I think 12 papers out of 53 papers (22.6%) in the references list are coming from one of the co-author papers, i.e. Wu, S. I am not sure what is the allowable percentage of self-citation in the research papers according to ISI. Thank you.
